# State-level prevalence estimates of latent tuberculosis infection in the United States by medical risk factors, demographic characteristics and nativity

**Ali Mirzazadeh**[1,2]*, **James G. Kahn**[2], **Maryam B. Haddad**[3], **Andrew N. Hill**[3], **Suzanne M. Marks**[3], **Adam Readhead**[4], **Pennan M. Barry**[4], **Jennifer Flood**[4], **Jonathan H. Mermin**[5], **Priya B. Shete**[2,6]

1 Department of Epidemiology & Biostatistics, Institute for Global Health Sciences, University of California San Francisco, San Francisco, California, United States of America, 2 Consortium to Assess Prevention Economics, Philip R Lee Institute for Health Policy Studies, University of California San Francisco, San Francisco, California, United States of America, 3 Division of Tuberculosis Elimination, National Center for HIV/AIDS, Viral Hepatitis, STD, and TB Prevention, US Centers for Disease Control and Prevention, Atlanta, Georgia, United States of America, 4 Division of Communicable Disease Control, California Department of Public Health, Tuberculosis Control Branch, Center for Infectious Diseases, Richmond, California, United States of America, 5 National Center for HIV/AIDS, Viral Hepatitis, STD, and TB Prevention, US Centers for Disease Control and Prevention, Atlanta, Georgia, United States of America, 6 Division of Pulmonary and Critical Care Medicine, University of California San Francisco and Zuckerberg San Francisco General Hospital, San Francisco, California, United States of America

* ali.mirzazadeh@ucsf.edu

## Abstract

### Introduction

Preventing tuberculosis (TB) disease requires treatment of latent TB infection (LTBI) as well as prevention of person-to-person transmission. We estimated the LTBI prevalence for the entire United States and for each state by medical risk factors, age, and race/ethnicity, both in the total population and stratified by nativity.

### Methods

We created a mathematical model using all incident TB disease cases during 2013–2017 reported to the National Tuberculosis Surveillance System that were classified using geno-type-based methods or imputation as not attributed to recent TB transmission. Using the annual average number of TB cases among US-born and non-US-born persons by medical risk factor, age group, and race/ethnicity, we applied population-specific reactivation rates (and corresponding 95% confidence intervals [CI]) to back-calculate the estimated prevalence of untreated LTBI in each population for the United States and for each of the 50 states and the District of Columbia in 2015.

### Results

We estimated that 2.7% (CI: 2.6%–2.8%) of the U.S. population, or 8.6 (CI: 8.3–8.8) million people, were living with LTBI in 2015. Estimated LTBI prevalence among US-born persons

**Data Availability Statement:** The data contain information abstracted from the national TB case report form called the Report of Verified Case of

Tuberculosis (RVCT) (OMB No. 0920-0026). These
data have been reported voluntarily to CDC by state
and local health departments, and are protected
under the Assurance of Confidentiality (Sections
306 and 308(d) of the Public Health Service Act, 42
U.S.C. 242k and 242m(d)), which prevents
disclosure of any information that could be used to
directly or indirectly identify patients. For more
information, see the CDC/ATSDR Policy on
Releasing and Sharing Data (at http://www.cdc.
gov/maso/Policy/ReleasingData.pdf). A limited
dataset is available at https://wonder.cdc.gov/tb.
html. Researchers seeking additional data may
apply to analyze National Tuberculosis Surveillance
System data at CDC headquarters by contacting
TBInfo@cdc.gov.

**Funding:** This project is supported by U.S. Centers
for Disease Control and Prevention (CDC), National
Center for HIV/AIDS, Viral Hepatitis, STD, and TB
Prevention Epidemiologic and Economic Modeling
Agreement (NEEMA) #U38PS004649. The findings
and conclusions in this report are those of the
authors and do not represent the official position of
the CDC. Based on the CDC mandate to collect
national infectious disease surveillance data, the
funder was involved in targeted data collection for
this study; however, authors had final say on which
data to include in the analysis.

**Competing interests:** The authors have declared
that no competing interests exist.

was 1.0% (CI: 1.0%–1.1%) and among non-US-born persons was 13.9% (CI: 13.5%–
14.3%). Among US-born persons, the highest LTBI prevalence was in persons aged ≥65
years (2.1%) and in persons of non-Hispanic Black race/ethnicity (3.1%). Among non-US-
born persons, the highest LTBI prevalence was estimated in persons aged 45–64 years
(16.3%) and persons of Asian and other racial/ethnic groups (19.1%).

## Conclusions

Our estimations of the prevalence of LTBI by medical risk factors and demographic charac-
teristics for each state could facilitate planning for testing and treatment interventions to
eliminate TB in the United States. Our back-calculation method feasibly estimates untreated
LTBI prevalence and can be updated using future TB disease case counts at the state or
national level.

## Introduction

Tuberculosis (TB) incidence in the United States has declined substantially over the past sev-
eral decades. In fact, the U.S. TB rate during 2019 declined to the lowest level on record, 27
cases per million persons (1.6% decline from 2018) [1] However, the rate has plateaued near
30 cases per million population annually since 2013 [2, 3]. The annual pace of decline remains
too slow to meet the national TB elimination goal of less than one case per million [4]. Geno-
typing of *Mycobacterium tuberculosis* isolates demonstrates that approximately 85% of TB dis-
ease cases in the United States are attributed to reactivation of latent infection with
*Mycobacterium tuberculosis* that was acquired >2 years prior [3].

Preventing TB requires treatment of *Mycobacterium tuberculosis* infection that might prog-
ress to TB disease [5]. Targeted testing and treatment is needed to prevent TB in the large res-
ervoir of persons with longstanding latent TB infection (LTBI) [6, 7]. Estimations of the
prevalence of untreated LTBI in populations at risk for TB by state could facilitate planning for
testing and treatment interventions to accelerate the TB decline and eliminate TB in the
United States.

Unfortunately, estimating the true burden of LTBI prevalence is challenging because LTBI
is not a reportable condition in most U.S. states. A study of data from the National Health and
Nutrition Examination Survey (NHANES) 2011–2012, which was the most recent cycle to test
for TB infection, estimated that approximately 13.3 (95% CI: 9.6–17.8) million noninstitution-
alized civilian U.S. residents would have a positive tuberculin skin test for TB infection [8].
Similar testing-based estimates at the state or local level are unavailable, because implementing
a representative population-based prevalence survey would be too time- and resource-inten-
sive for most state health authorities. However, relying on national estimates to inform state or
local programs and policies for targeted testing and treatment of LTBI may potentially leading
to wasted resources if, in reality, local estimates and populations at risk differ from the national
pattern.

Recently, Haddad *et al.* [9] estimated untreated LTBI prevalence at the state and county
level using a uniform annual reactivation rate applied to the entire population. However, their
methodology did not estimate LTBI prevalence within populations having medical risk factors
that increase risk for TB progression, nor stratify LTBI prevalence by demographic characteris-
tics such as age or race/ethnicity. Having more detailed state-level estimates could be

informative for identifying those populations that would most benefit from TB preventive care strategies. Several TB models demonstrate the potential health impact and attractive cost-effectiveness of expanded testing and treatment for LTBI in populations at high risk for TB [6, 10]. Without treatment, patients with LTBI have a 5%–10% lifetime risk of progression to TB [11, 12]; that risk, however, varies based on individual medical risk factors and certain demographic characteristics such as age.

In this study, we applied previously derived population-specific annual reactivation rates to population-specific annual average counts of TB cases to back-calculate estimates of LTBI prevalence for the entire United States, as well as for each of the 50 U.S. states and the District of Columbia, by medical risk factor, age group, and race/ethnicity, both in the total population and stratified by nativity (i.e., US-born or non-US–born), in 2015.

## Methods

### Data source: Counts of reported TB cases not attributed to recent transmission

The National Tuberculosis Surveillance System [2, 3] provided aggregate counts of reported cases of TB disease in the 50 U.S. states and the District of Columbia during 2013–2017 that were not attributed to recent transmission. In the United States, recent transmission is now routinely estimated using the France *et al.* field-validated plausible source-case method [2, 3, 13] (i.e., plausible infectious source case in a person ≥10 years of age within 10 miles in the previous 2 years having a matching genotype result). We back-calculated solely from those TB cases not attributed or imputed (see below for details) to recent transmission because our focus was on estimating longstanding LTBI that could be diagnosed by targeted testing and then treated. For similar reasons, we excluded all cases occurring in children under the age of 1 year.

### Back-calculation method overview

Our back-calculation method applied previously derived population-specific TB reactivation rates from the literature [14, 15]. We divided the average annual count of TB cases not attributed to recent transmission by the corresponding estimated TB reactivation rates (point estimates and 95% confidence intervals [CI]) to estimate counts of people with those characteristics who were living with LTBI in 2015. We repeated this calculation for US-born and non-US–born populations iteratively classified into three groupings based on five medical risk factors, five age groups, and four race/ethnicity categories reported to the National Tuberculosis Surveillance System. Reported medical risk factors among TB cases were used to estimate medical risk factors among persons with LTBI.

Because the total count of persons estimated to have LTBI when summed across medical risk factor, age, and race/ethnicity categories differed from the total estimated LTBI count, we considered the sum across the five age groups within the stratified US-born and non-US–born populations to be the referent total. We then proportionally adjusted the estimated LTBI counts within the medical risk factor and race/ethnicity categories to match that referent.

Finally, to provide LTBI estimates as a proportion of the underlying population, we used the 2015 American Community Survey midpoint estimates [16] for each state's population size by age group, race/ethnicity, and nativity. Similar state-level denominators for medical risk factor prevalence stratified by nativity are not available, so those proportions are not presented.

For this analysis, we used freely available R 3.6.3 software (R Core Team, Vienna, Austria) [17]. The R code for the back-calculation model, which can be adapted for any jurisdiction or time period, is included as **S2 Appendix in** S1 File.

## Model inputs: Estimated annual TB reactivation rates among people living with LTBI

Our back-calculation model inputs included the previously derived TB reactivation rates in the United States reported by Shea *et al.* [14] and the reactivation rate ratios (RRR) from an international systematic review conducted by Yeats [15]. Similar to our analysis, the Shea *et al.* estimates excluded children under the age of 1 year and, using an earlier genotype-based methodology, TB cases attributed to recent transmission. Shea *et al.* also stratified TB reactivation rates by nativity [14]. The Yeats systematic review provided the RRRs used to derive TB reactivation rates for all persons, regardless of nativity, with certain medical risk factors [15]. The estimated reactivation rates for all population groupings are presented in Table 1.

Shea *et al.* [14] estimated an overall TB reactivation rate of 0.084 (95% CI 0.083–0.085) per 100 person-years in the total population. By nativity, the estimated reactivation rate was 0.082 (0.080–0.083) among US-born and 0.098 (0.096–0.100) among non-US–born persons. Estimated reactivation rates were higher among people living with HIV (1.82) and varied across

**Table 1. Back-calculation model inputs: Estimated reactivation rates (per 100 person-years) by medical risk factors and demographic characteristics, stratified by nativity (U.S. birth or non-US birth).**

| Groupings | US-born (95% confidence interval*) | non-US–born (95% confidence interval*) |
|---|---|---|
| **Medical risk factors*** | | |
| With solid organ transplant (SOT) | 2.616 (2.024, 3.345) | 3.126 (2.429, 4.030) |
| With HIV coinfection but not SOT | 1.468 (1.048, 1.519) | 1.754 (1.258, 1.830) |
| With end-stage renal disease (ESRD) but not SOT or HIV coinfection | 0.932 (0.576, 1.502) | 1.114 (0.691, 1.810) |
| With immunosuppressive therapy (e.g., TNF-alpha blocker use) but not SOT, HIV coinfection, or ESRD | 0.382 (0.196, 0.439) | 0.457 (0.235, 0.530) |
| With diabetes, but not SOT, HIV coinfection, ESRD, or immunosuppressive therapy (e.g., TNF-alpha blocker use) | 0.169 (0.125, 0.225) | 0.202 (0.149, 0.272) |
| None of the above medical risk factors reported (i.e., overall population estimates*) | 0.082 (0.080, 0.083) | 0.098 (0.096, 0.100) |
| Age groups** | | |
| 1–14 years | 0.180 (0.165, 0.192) | 0.062 (0.057, 0.068) |
| 15–24 years | 0.110 (0.102, 0.118) | 0.156 (0.150, 0.162) |
| 25–44 years | 0.096 (0.092, 0.100) | 0.080 (0.078, 0.082) |
| 45–64 years | 0.064 (0.062, 0.066) | 0.074 (0.072, 0.076) |
| 65+ years | 0.072 (0.069, 0.074) | 0.256 (0.248, 0.265) |
| Race/Ethnicity** | | |
| Non-Hispanic White | 0.067 (0.065, 0.069) | 0.023 (0.021, 0.024) |
| Non-Hispanic Black | 0.075 (0.073, 0.077) | 0.200 (0.193, 0.208) |
| Hispanic | 0.178 (0.171, 0.186) | 0.086 (0.084, 0.088) |
| Asian/other | 0.141 (0.131, 0.151) | 0.138 (0.135, 0.141) |

* The point estimates and 95% confidence intervals are derived from the previously derived reactivation rate ratios reported by Yeats *et al.* multiplied by the reactivation rates reported for the overall US-born and non-US–born population, without regard to medical risk factor presence, in Shea *et al.* paper. See **S1 Appendix in** S1 File for more details.

** The reactivation rates and 95% confidence intervals by age group and race/ethnicity, without regard to medical risk factor presence, are taken from Shea *et al.* paper. TNF: tumor necrosis factor; US: United States.

age (range 0.064 to 0.112) and race/ethnicity (range 0.045 to 0.190) [14]. Because the Shea *et al.* estimates grouped together all children aged 1–14 years, our estimates do the same. Our model also incorporated the Shea *et al.* rates for the remaining age groups (15–24, 25–44, 45–64, ≥65 years), and used the same 4 race/ethnicity groupings (non-Hispanic White, non-Hispanic Black, Hispanic, Asian/other). The fourth race/ethnicity category, Asian/other, is a combination of American Indian/Alaska Native, Asian, Native Hawaiian/other Pacific Islander, and multiple race.

To estimate reactivation rates among persons with certain medical risk factors (which, except for HIV, were not available in Shea *et al.*), we multiplied the Shea *et al.* [14] overall US-born and non-US–born reactivation rates, without regard to medical risk factor presence, by the Yeats (11) medical risk RRRs (**S1 Appendix in** S1 File). The Yeats RRRs by medical risk factor were 31.90 (25.30–40.30) for solid organ transplant, 17.90 (13.10–18.30) for HIV, 11.37 (7.20–18.10) for end-stage renal disease (ESRD), 4.66 (2.45–5.30) for immunosuppressive therapy, and 2.06 (1.56–2.72) for diabetes.

### Approach for missing data and TB cases with >1 medical risk factor

All TB cases lacking documentation of *M. tuberculosis* culture positivity (and thus genotype result) were missing a recent transmission determination. Missing age (n = 7 cases), nativity (n = 34), race/ethnicity (n = 105), and recent transmission (n = 12,249) variables were imputed using predictive mean matching with state, reporting year, and medical risk factors as covariates in the imputation model (R "mice" package) [18]. We used multiple imputation (5 runs) with a random seed to impute missing data. Separate imputations were conducted for US-born and non-US–born subgroups. TB cases with multiple medical risk factors were hierarchically classified into the risk factor category having the highest reactivation rate.

## Results

### Estimated U.S. population prevalence of LTBI

The estimated number of people with untreated LTBI in the United States in 2015 was 8,561,899 (95% CI 8,307,006–8,844,338) (Table 2) of which 2,716,529 (32%) were US-born and 5,845,369 (68%) were non-US–born persons. The estimated total population untreated LTBI prevalence in the United States was 2.7% (2.7%–2.8%) (Table 3).

Estimated prevalence among non-US–born persons was 14 times the estimated prevalence among US-born persons (13.9% vs. 1.0%) (Table 3). National LTBI prevalence estimates ranged from 0.2% for children aged 1–14 years to 4.2% for adults aged 45–64 years (Table 4). By race/ethnicity, persons of Asian or other race/ethnicity in the total U.S. population were estimated to have the highest LTBI prevalence (8.7%). Among US-born persons, those aged ≥65 years (2.1%) or non-Hispanic Black (3.1%) were estimated to have the highest LTBI prevalence. Among non-US–born persons, those aged 45–64 years (16.3%) or Asian or other race/ethnicity (19.1%) were estimated to have the highest LTBI prevalence.

### Characteristics of persons predicted to have LTBI

About 10.5% of the US-born persons estimated as having LTBI had medical risk factors (Table 2); the most common risk factor (8.0%) was diabetes without other concomitant conditions such as ESRD or HIV, followed by immunosuppressive therapy (1.7%). Most of the US-born persons estimated as having LTBI were non-Hispanic White (46.5%). The predominant age group was 45–64 years (44.2%).

**Table 2. Estimated number of people living with latent tuberculosis infection in the United States in 2015, by medical risk factors and demographic characteristics, stratified by nativity.**

| Groupings | US-born (95% CI*) | | Non-US–born (95% CI*) | | Total (95% CI*) | |
|---|---|---|---|---|---|---|
| **Total** | 2,716,529 (2,625,156 to 2,828,853) | 100.0% | 5,845,369 (5,681,850 to 6,015,484) | 100.0% | 8,561,899 (8,307,006 to 8,844,338) | 100.0% |
| **Medical risk factors** | | | | | | |
| With solid organ transplant (SOT) | 681 (658 to 709) | <0.03% | 1,236 (1,201 to 1,271) | <0.03% | 1,917 (1,859 to 1,981) | <0.03% |
| With HIV coinfection but not SOT | 12,744 (12,316 to 13,271) | 0.5% | 17,657 (17,164 to 18,171) | 0.3% | 30,402 (29,479 to 31,442) | 0.4% |
| With end-stage renal disease (ESRD) but not SOT or HIV coinfection | 7,955 (7,687 to 8,284) | 0.3% | 14,788 (14,374 to 15,218) | 0.3% | 22,743 (22,062 to 23,503) | 0.3% |
| With immunosuppressive therapy (e.g., TNF-alpha blocker users) but not SOT or HIV coinfection or ESRD | 45,308 (43,783 to 47,182) | 1.7% | 61,466 (59,746 to 63,255) | 1.1% | 106,774 (103,529 to 110,437) | 1.2% |
| With diabetes but not SOT or HIV coinfection or ESRD or immunosuppressive therapy (TNF-alpha blocker users) | 217,372 (210,060 to 226,361) | 8.0% | 626,100 (608,584 to 644,321) | 10.7% | 843,471 (818,644 to 870,682) | 9.9% |
| None of the above medical risk factors reported | 2,432,469 (2,350,652 to 2,533,046) | 89.5% | 5,124,123 (4,980,781 to 5,273,247) | 87.7% | 7,556,592 (7,331,433 to 7,806,292) | 88.3% |
| **Age groups** | | | | | | |
| 1–14 years | 39,689 (37,208 to 43,297) | 1.5% | 99,548 (90,765 to 108,281) | 1.7% | 139,237 (127,973 to 151,578) | 1.6% |
| 15–24 years | 166,000 (154,746 to 179,020) | 6.1% | 379,179 (365,136 to 394,347) | 6.5% | 545,179 (519,882 to 573,366) | 6.4% |
| 25–44 years | 450,500 (432,480 to 470,087) | 16.6% | 2,522,750 (2,461,220 to 2,587,436) | 43.2% | 2,973,250 (2,893,700 to 3,057,523) | 34.7% |
| 45–64 years | 1,200,562 (1,164,182 to 1,239,290) | 44.2% | 2,278,595 (2,218,632 to 2,341,889) | 39.0% | 3,479,157 (3,382,813 to 3,581,179) | 40.6% |
| 65+ years | 859,778 (836,541 to 897,159) | 31.6% | 565,297 (546,098 to 583,532) | 9.7% | 1,425,075 (1,382,639 to 1,480,692) | 16.6% |
| **Race/Ethnicity** | | | | | | |
| Non-Hispanic White | 1,263,101 (1,220,621 to 1,315,319) | 46.5% | 1,165,940 (1,133,327 to 1,199,869) | 19.9% | 2,429,041 (2,353,947 to 2,515,189) | 28.4% |
| Non-Hispanic Black | 1,075,748 (1,039,563 to 1,120,229) | 39.6% | 408,514 (397,086 to 420,403) | 7.0% | 1,484,262 (1,436,650 to 1,540,632) | 17.3% |
| Hispanic | 232,730 (224,899 to 242,358) | 8.6% | 2,144,785 (2,084,784 to 2,207,207) | 36.7% | 2,377,515 (2,309,683 to 2,449,565) | 27.8% |
| Asian/other | 144,950 (140,073 to 150,947) | 5.3% | 2,126,130 (2,066,653 to 2,188,006) | 36.4% | 2,271,080 (2,206,726 to 2,338,952) | 26.5% |

* CI = confidence interval; 95% CI based solely on previously derived population-specific reactivation rates (see Table 1 and **S1 Appendix in** S1 File). For example, the reactivation rates for medical risk factors were calculated by multiplying the reactivation rate ratios reported by Yeats et al. with the reactivation rates reported for the overall US-born and non-US–born populations, without regard to medical risk factor presence, in the Shea *et al.* paper

About 12.4% of the non-US–born persons estimated as having LTBI had medical risk factors. The most common (10.7%) was diabetes without other concomitant conditions such as ESRD or HIV, followed by immunosuppressive therapy (1.1%). Most of the non-US–born persons estimated as having LTBI were Hispanic (36.7%) or of Asian or other race/ethnicity (36.4%). The predominant age group was 25–44 years (43.2%).

## State-level estimates of LTBI prevalence

The 2 states estimated to have the highest number of persons living with LTBI were California (1,722,575) and Texas (1,081,749). Estimated total population LTBI prevalence in the 4 states (California, New York, Texas, Florida) with the highest annual counts of TB cases ranged from 3.0% in Florida to 4.5% in California. In 11 states, estimated total population LTBI prevalence

**Table 3. Estimated number and proportion of persons living with latent tuberculosis infection in 2015, stratified by nativity and in total population, in 50 U.S. states and District of Columbia.**

| Area | US-born Persons (95% CI*) | Non-US–born Persons (95% CI*) | Total Persons (95% CI*) | US-born % (95% CI*) | Non-US–born % (95% CI*) | Total % (95% CI*) | % Non-US–born out of the total |
|---|---|---|---|---|---|---|---|
| **US** | 2,716,529 (2,625,156 to 2,828,853) | 5,845,369 (5,681,850 to 6,015,484) | 8,561,899 (8,307,006 to 8,844,338) | 1.0 (1.0 to 1.0) | 13.9 (13.5 to 14.3) | 2.7 (2.7 to 2.8) | 68% |
| AK | 22,085 (21,367 to 22,929) | 10,516 (10,221 to 10,821) | 32,601 (31,588 to 33,750) | 3.5 (3.4 to 3.7) | 17.8 (17.3 to 18.3) | 4.8 (4.6 to 4.9) | 32% |
| AL | 83,952 (81,326 to 87,243) | 25,393 (24,723 to 26,097) | 109,345 (106,049 to 113,340) | 1.8 (1.8 to 1.9) | 15.6 (15.2 to 16.0) | 2.3 (2.3 to 2.4) | 23% |
| AR | 56,812 (54,993 to 59,147) | 29,062 (28,130 to 30,032) | 85,874 (83,123 to 89,179) | 2.1 (2.0 to 2.2) | 17.3 (16.7 to 17.8) | 3.0 (2.9 to 3.1) | 34% |
| AZ | 55,854 (53,969 to 58,233) | 130,712 (126,946 to 134,635) | 186,566 (180,916 to 192,868) | 1.0 (1.0 to 1.1) | 12.2 (11.8 to 12.5) | 2.8 (2.8 to 2.9) | 70% |
| CA | 321,222 (309,143 to 335,790) | 1,401,353 (1,362,491 to 1,441,551) | 1,722,575 (1,671,633 to 1,777,341) | 1.1 (1.1 to 1.2) | 13.9 (13.5 to 14.3) | 4.5 (4.4 to 4.6) | 81% |
| CO | 16,660 (16,123 to 17,342) | 53,272 (51,803 to 54,797) | 69,932 (67,926 to 72,139) | 0.4 (0.3 to 0.4) | 8.7 (8.5 to 9.0) | 1.3 (1.3 to 1.4) | 76% |
| CT | 13,085 (12,651 to 13,642) | 48,870 (47,569 to 50,230) | 61,955 (60,220 to 63,872) | 0.4 (0.4 to 0.4) | 10.1 (9.8 to 10.3) | 1.8 (1.7 to 1.8) | 79% |
| DC | 10,190 (9,871 to 10,616) | 18,998 (18,505 to 19,517) | 29,188 (28,376 to 30,133) | 1.9 (1.8 to 2.0) | 18.3 (17.8 to 18.8) | 4.5 (4.4 to 4.6) | 65% |
| DE | 7,855 (7,597 to 8,185) | 11,579 (11,257 to 11,916) | 19,434 (18,854 to 20,101) | 0.9 (0.9 to 1.0) | 12.7 (12.3 to 13.1) | 2.1 (2.0 to 2.2) | 60% |
| FL | 231,286 (223,679 to 240,440) | 355,927 (346,105 to 366,176) | 587,214 (569,784 to 606,616) | 1.5 (1.4 to 1.5) | 9.1 (8.9 to 9.4) | 3.0 (2.9 to 3.1) | 61% |
| GA | 135,629 (131,087 to 141,076) | 158,030 (153,601 to 162,649) | 293,659 (284,689 to 303,725) | 1.5 (1.5 to 1.6) | 15.8 (15.4 to 16.3) | 3.0 (2.9 to 3.1) | 54% |
| HI | 16,721 (16,117 to 17,500) | 86,267 (83,793 to 88,832) | 102,989 (99,910 to 106,332) | 1.5 (1.4 to 1.6) | 37.6 (36.5 to 38.7) | 7.6 (7.4 to 7.9) | 84% |
| IA | 12,422 (12,007 to 12,919) | 37,322 (36,317 to 38,375) | 49,744 (48,325 to 51,294) | 0.4 (0.4 to 0.4) | 24.4 (23.7 to 25.1) | 1.6 (1.6 to 1.7) | 75% |
| ID | 2,917 (2,817 to 3,049) | 7,946 (7,675 to 8,226) | 10,863 (10,492 to 11,275) | 0.2 (0.2 to 0.2) | 7.3 (7.0 to 7.5) | 0.7 (0.7 to 0.7) | 73% |
| IL | 90,741 (87,688 to 94,480) | 215,059 (208,999 to 221,352) | 305,800 (296,687 to 315,832) | 0.8 (0.8 to 0.9) | 12.4 (12.1 to 12.8) | 2.4 (2.4 to 2.5) | 70% |
| IN | 42,699 (41,310 to 44,418) | 62,741 (60,969 to 64,596) | 105,440 (102,278 to 109,014) | 0.7 (0.7 to 0.7) | 16.5 (16.0 to 17.0) | 1.7 (1.6 to 1.7) | 60% |
| KS | 10,646 (10,299 to 11,084) | 26,162 (25,409 to 26,949) | 36,808 (35,708 to 38,033) | 0.4 (0.4 to 0.4) | 10.6 (10.3 to 10.9) | 1.3 (1.3 to 1.4) | 71% |
| KY | 42,148 (40,837 to 43,818) | 36,081 (35,003 to 37,203) | 78,229 (75,839 to 81,021) | 1.0 (1.0 to 1.1) | 19.4 (18.8 to 20.0) | 1.8 (1.8 to 1.9) | 46% |
| LA | 83,104 (80,366 to 86,330) | 41,461 (40,302 to 42,671) | 124,565 (120,668 to 129,001) | 1.9 (1.9 to 2.0) | 19.2 (18.6 to 19.7) | 2.8 (2.7 to 2.9) | 33% |
| MA | 30,362 (29,282 to 31,706) | 160,542 (156,032 to 165,236) | 190,904 (185,314 to 196,942) | 0.6 (0.5 to 0.6) | 14.5 (14.1 to 14.9) | 2.9 (2.8 to 3.0) | 84% |
| MD | 38,985 (37,726 to 40,548) | 153,987 (149,724 to 158,441) | 192,972 (187,451 to 198,989) | 0.8 (0.8 to 0.8) | 16.8 (16.3 to 17.3) | 3.3 (3.2 to 3.4) | 80% |
| ME | 4,951 (4,806 to 5,146) | 12,350 (11,936 to 12,778) | 17,302 (16,742 to 17,925) | 0.4 (0.4 to 0.4) | 28.6 (27.6 to 29.6) | 1.3 (1.3 to 1.4) | 71% |
| MI | 54,179 (52,456 to 56,372) | 76,644 (74,437 to 78,935) | 130,823 (126,893 to 135,307) | 0.6 (0.6 to 0.6) | 10.7 (10.4 to 11.0) | 1.3 (1.3 to 1.4) | 59% |
| MN | 20,270 (19,520 to 21,211) | 124,859 (121,085 to 128,784) | 145,129 (140,605 to 149,996) | 0.4 (0.4 to 0.4) | 25.6 (24.8 to 26.4) | 2.7 (2.6 to 2.8) | 86% |

*(Continued)*

**Table 3.** (*Continued*)

| Area | US-born | Non-US–born | Total | US-born | Non-US–born | Total | % Non-US–born out of the total |
|---|---|---|---|---|---|---|---|
| | Persons (95% CI*) | Persons (95% CI*) | Persons (95% CI*) | % (95% CI*) | % (95% CI*) | % (95% CI*) | |
| MO | 42,708 (41,363 to 44,443) | 50,626 (49,205 to 52,115) | 93,334 (90,568 to 96,559) | 0.7 (0.7 to 0.8) | 29.4 (28.5 to 30.2) | 1.6 (1.5 to 1.6) | 54% |
| MS | 55,646 (53,928 to 57,826) | 11,784 (11,451 to 12,131) | 67,430 (65,379 to 69,957) | 2.0 (1.9 to 2.0) | 13.0 (12.6 to 13.4) | 2.3 (2.2 to 2.4) | 17% |
| MT | 6,562 (6,369 to 6,820) | 506 (491 to 523) | 7,069 (6,860 to 7,344) | 0.7 (0.7 to 0.7) | 3.3 (3.2 to 3.4) | 0.7 (0.7 to 0.7) | 7% |
| NC | 101,201 (97,902 to 105,297) | 102,953 (100,093 to 105,936) | 204,154 (197,995 to 211,233) | 1.1 (1.1 to 1.2) | 12.7 (12.3 to 13.0) | 2.1 (2.0 to 2.2) | 50% |
| ND | 5,814 (5,606 to 6,065) | 8,996 (8,713 to 9,290) | 14,809 (14,319 to 15,355) | 0.8 (0.8 to 0.9) | 29.7 (28.7 to 30.7) | 2.1 (2.0 to 2.1) | 61% |
| NE | 5,961 (5,750 to 6,238) | 23,159 (22,526 to 23,824) | 29,120 (28,276 to 30,062) | 0.3 (0.3 to 0.4) | 16.8 (16.4 to 17.3) | 1.6 (1.5 to 1.6) | 80% |
| NH | 2,779 (2,699 to 2,888) | 12,911 (12,530 to 13,307) | 15,690 (15,229 to 16,196) | 0.2 (0.2 to 0.2) | 17.1 (16.6 to 17.6) | 1.2 (1.2 to 1.2) | 82% |
| NJ | 51,609 (49,915 to 53,804) | 238,214 (231,668 to 245,043) | 289,823 (281,583 to 298,846) | 0.7 (0.7 to 0.8) | 12.1 (11.7 to 12.4) | 3.3 (3.2 to 3.4) | 82% |
| NM | 22,340 (21,655 to 23,219) | 25,799 (25,074 to 26,550) | 48,139 (46,729 to 49,769) | 1.2 (1.2 to 1.3) | 12.9 (12.5 to 13.3) | 2.4 (2.3 to 2.5) | 54% |
| NV | 18,415 (17,765 to 19,196) | 52,964 (51,451 to 54,533) | 71,379 (69,217 to 73,729) | 0.8 (0.8 to 0.8) | 10.3 (10.0 to 10.6) | 2.6 (2.5 to 2.6) | 74% |
| NY** | 116,927 (112,912 to 121,895) | 589,623 (573,346 to 606,588) | 706,551 (686,257 to 728,482) | 0.8 (0.7 to 0.8) | 13.4 (13.0 to 13.8) | 3.6 (3.5 to 3.7) | 83% |
| OH | 61,670 (59,722 to 64,175) | 89,057 (86,452 to 91,766) | 150,727 (146,174 to 155,941) | 0.6 (0.5 to 0.6) | 17.6 (17.1 to 18.1) | 1.3 (1.3 to 1.4) | 59% |
| OK | 37,719 (36,473 to 39,270) | 25,802 (24,962 to 26,672) | 63,521 (61,435 to 65,942) | 1.1 (1.1 to 1.1) | 11.4 (11.0 to 11.7) | 1.7 (1.7 to 1.8) | 41% |
| OR | 19,374 (18,724 to 20,171) | 48,843 (47,446 to 50,293) | 68,216 (66,171 to 70,465) | 0.6 (0.5 to 0.6) | 12.1 (11.7 to 12.4) | 1.7 (1.7 to 1.8) | 72% |
| PA | 79,760 (77,253 to 82,930) | 124,124 (120,687 to 127,707) | 203,884 (197,940 to 210,636) | 0.7 (0.7 to 0.7) | 15.3 (14.9 to 15.8) | 1.6 (1.6 to 1.7) | 61% |
| RI | 3,997 (3,872 to 4,161) | 15,199 (14,764 to 15,650) | 19,196 (18,636 to 19,811) | 0.4 (0.4 to 0.5) | 10.3 (10.0 to 10.6) | 1.9 (1.8 to 1.9) | 79% |
| SC | 77,487 (75,010 to 80,577) | 24,873 (24,187 to 25,593) | 102,360 (99,197 to 106,169) | 1.7 (1.7 to 1.8) | 12.6 (12.3 to 13.0) | 2.2 (2.1 to 2.3) | 24% |
| SD | 6,553 (6,337 to 6,809) | 5,386 (5,244 to 5,535) | 11,939 (11,581 to 12,344) | 0.8 (0.8 to 0.8) | 32.7 (31.8 to 33.6) | 1.4 (1.4 to 1.5) | 45% |
| TN | 69,647 (67,457 to 72,467) | 52,980 (51,516 to 54,512) | 122,627 (118,973 to 126,979) | 1.1 (1.1 to 1.2) | 15.0 (14.6 to 15.4) | 1.9 (1.9 to 2.0) | 43% |
| TX | 402,891 (388,859 to 419,553) | 678,859 (660,065 to 698,479) | 1,081,749 (1,048,925 to 1,118,032) | 1.8 (1.8 to 1.9) | 15.5 (15.0 to 15.9) | 4.1 (4.0 to 4.2) | 63% |
| UT | 4,725 (4,557 to 4,931) | 23,782 (23,074 to 24,517) | 28,507 (27,630 to 29,448) | 0.2 (0.2 to 0.2) | 10.4 (10.1 to 10.7) | 1.0 (1.0 to 1.0) | 83% |
| VA | 43,992 (42,541 to 45,865) | 161,036 (156,438 to 165,818) | 205,027 (198,979 to 211,683) | 0.6 (0.6 to 0.6) | 16.7 (16.2 to 17.2) | 2.5 (2.4 to 2.6) | 79% |
| VT | 1,736 (1,682 to 1,798) | 3,625 (3,532 to 3,723) | 5,361 (5,214 to 5,521) | 0.3 (0.3 to 0.3) | 12.5 (12.2 to 12.8) | 0.9 (0.9 to 0.9) | 68% |
| WA | 39,194 (37,770 to 40,896) | 152,217 (148,013 to 156,595) | 191,411 (185,783 to 197,491) | 0.6 (0.6 to 0.7) | 16.7 (16.2 to 17.2) | 2.7 (2.7 to 2.8) | 80% |
| WI | 20,102 (19,433 to 20,903) | 31,569 (30,707 to 32,466) | 51,671 (50,141 to 53,369) | 0.4 (0.4 to 0.4) | 12.2 (11.9 to 12.6) | 0.9 (0.9 to 0.9) | 61% |

(*Continued*)

**Table 3.** (Continued)

| Area | US-born | Non-US–born | Total | US-born | Non-US–born | Total | % Non-US–born out of the total |
|---|---|---|---|---|---|---|---|
| | Persons (95% CI*) | Persons (95% CI*) | Persons (95% CI*) | % (95% CI*) | % (95% CI*) | % (95% CI*) | |
| WV | 11,799 (11,449 to 12,239) | 4,198 (4,064 to 4,337) | 15,997 (15,512 to 16,577) | 0.7 (0.6 to 0.7) | 18.9 (18.3 to 19.5) | 0.9 (0.9 to 0.9) | 26% |
| WY | 1,146 (1,114 to 1,192) | 1,148 (1,118 to 1,180) | 2,294 (2,232 to 2,372) | 0.2 (0.2 to 0.2) | 6.1 (6.0 to 6.3) | 0.4 (0.4 to 0.4) | 50% |

*CI = confidence interval; 95% CI based solely on previously derived population-specific reactivation rates (see Table 1 and **S1 Appendix in** S1 File).
**New York city and NY (rest of NY) are combined when producing NY estimates.

was ≥3%. In 19 states, the estimated LTBI prevalence in the US-born population was ≥1%. In 26 states, the estimated LTBI prevalence in the non-US–born population was ≥15% (Fig 1 and Table 3). The state-level LTBI prevalence by age groups and race/ethnicity are presented in **S1 and S2 Figs in** S1 File.

Each state's predicted total number of people with LTBI, as well as grouped by medical risk factor, age group, and race/ethnicity, is presented in S1 Table.

Median estimated state-level total population LTBI prevalence was 2.4% (interquartile range [IQR] 1.1%–4.2%). Median estimated state-level LTBI prevalence among US-born persons was 0.6% (IQR 0.3%–1.5%) and among non-US–born persons was 13.5%

**Table 4. Estimated prevalence of latent tuberculosis infection in the United States in 2015 within age group and race/ethnicity, stratified by nativity and in total population.**

| Groupings | US-born | | | Non-US–born | | | Total | | |
|---|---|---|---|---|---|---|---|---|---|
| | Pop. size (N) | No. with LTBI (95%CI) | % with LTBI (95% CI) | Pop. size (N) | No. with LTBI (95%CI) | % with LTBI (95% CI) | Pop. size (N) | No. with LTBI (95%CI) | % with LTBI (95% CI) |
| 1–14 years | 55,495,781 | 39,689 (37,208 to 43,297) | 0.1 (0.1 to 0.1) | 1,611,890 | 99,548 (90,765 to 108,281) | 6.2 (5.6 to 6.7) | 57,107,672 | 139,237 (127,973 to 151,578) | 0.2 (0.2 to 0.3) |
| 15–24 years | 39,112,317 | 166,000 (154,746 to 179,020) | 0.4 (0.4 to 0.5) | 3,943,054 | 379,179 (365,136 to 394,347) | 9.6 (9.3 to 10.0) | 43,055,372 | 545,179 (519,882 to 573,366) | 1.3 (1.2 to 1.3) |
| 25–44 years | 66,037,572 | 450,500 (432,480 to 470,087) | 0.7 (0.7 to 0.7) | 16,887,403 | 2,522,750 (2,461,220 to 2,587,436) | 14.9 (14.6 to 15.3) | 82,924,976 | 2,973,250 (2,893,700 to 3,057,523) | 3.6 (3.5 to 3.7) |
| 45–64 years | 69,263,607 | 1,200,562 (1,164,182 to 1,239,290) | 1.7 (1.7 to 1.8) | 13,949,859 | 2,278,595 (2,218,632 to 2,341,889) | 16.3 (15.9 to 16.8) | 83,213,466 | 3,479,157 (3,382,813 to 3,581,179) | 4.2 (4.1 to 4.3) |
| 65+ years | 40,218,195 | 859,778 (836,541 to 897,159) | 2.1 (2.1 to 2.2) | 5,775,767 | 565,297 (546,098 to 583,532) | 9.8 (9.5 to 10.1) | 45,993,962 | 1,425,075 (1,382,639 to 1,480,692) | 3.1 (3.0 to 3.2) |
| Non-Hispanic White | 185,609,880 | 1,263,101 (1,220,621 to 1,315,319) | 0.7 (0.7 to 0.7) | 7,794,914 | 1,165,940 (1,133,327 to 1,199,869) | 15.0 (14.5 to 15.4) | 193,404,794 | 2,429,041 (2,353,947 to 2,515,189) | 1.3 (1.2 to 1.3) |
| Non-Hispanic Black | 34,403,400 | 1,075,748 (1,039,563 to 1,120,229) | 3.1 (3.0 to 3.3) | 3,665,468 | 408,514 (397,086 to 420,403) | 11.1 (10.8 to 11.5) | 38,068,868 | 1,484,262 (1,436,650 to 1,540,632) | 3.9 (3.8 to 4.0) |
| Hispanic | 35,046,799 | 232,730 (224,899 to 242,358) | 0.7 (0.6 to 0.7) | 19,566,449 | 2,144,785 (2,084,784 to 2,207,207) | 11.0 (10.7 to 11.3) | 54,613,248 | 2,377,515 (2,309,683 to 2,449,565) | 4.4 (4.2 to 4.5) |
| Asian/other | 15,067,394 | 144,950 (140,073 to 150,947) | 1.0 (0.9 to 1.0) | 11,141,144 | 2,126,130 (2,066,653 to 2,188,006) | 19.1 (18.5 to 19.6) | 26,208,537 | 2,271,080 (2,206,726 to 2,338,952) | 8.7 (8.4 to 8.9) |

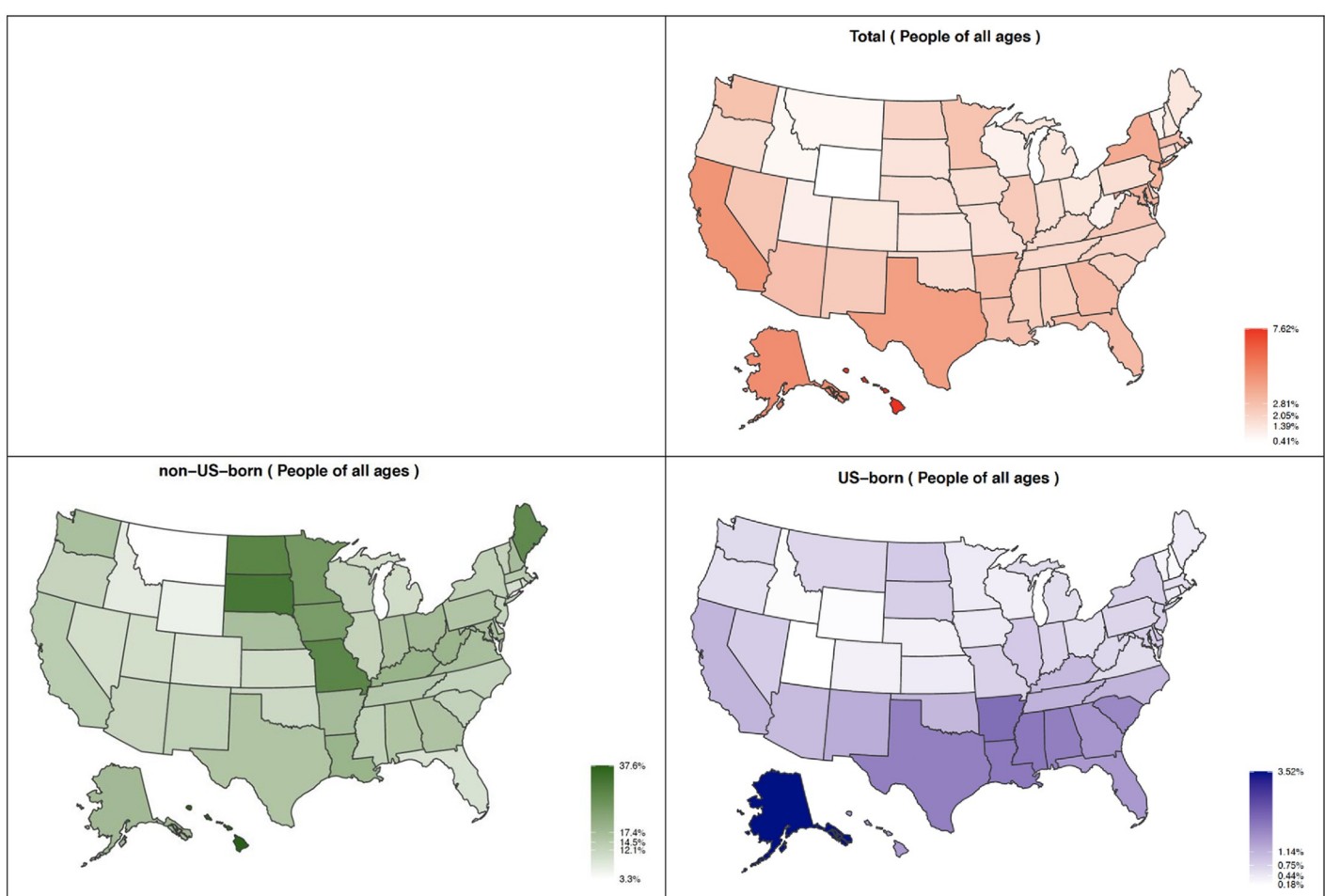

**Fig 1. Estimated prevalence of latent tuberculosis infection by state and nativity (US-born or non-US–born), United States, 2015.** The numbers in the legend are minimum, 25%, 50%, 75% quintiles, and maximum values. Software and source: open-source R and "usmap" package were used to create the maps. Both R software and the "usmap" package are license under GPL-3 | file LICENSE and free to use. [https://www.r-project.org/Licenses/; https://cran.r-project.org/web/packages/usmap/usmap.pdf].

(IQR 9.7%–18.9%). State-level estimates for the prevalence of LTBI by age group and race/ethnicity are presented in S1 Table and summarized visually in Fig 2. By race/ethnicity, highest median estimated state-level LTBI prevalence was among Asian or other race/ethnic groups (median 7.6%, IQR 5.4%–9.8%).

## Discussion

We used a back-calculation method using data from the National Tuberculosis Surveillance System, estimation and imputation of recent TB transmission, and previously published TB reactivation estimates to produce national and state-level estimates of LTBI prevalence, both in the total population and within sub-groups. Our model estimated 8.6 million people (2.7%) were living with untreated LTBI in the United States in 2015, of whom the majority (68%) were non-US–born. Estimated LTBI prevalence among US-born persons was 1.0% and among non-US–born persons was 13.9%. Among US-born persons, the highest LTBI prevalence was among persons aged ≥65 years (2.1%) and persons of non-Hispanic Black race/ethnicity

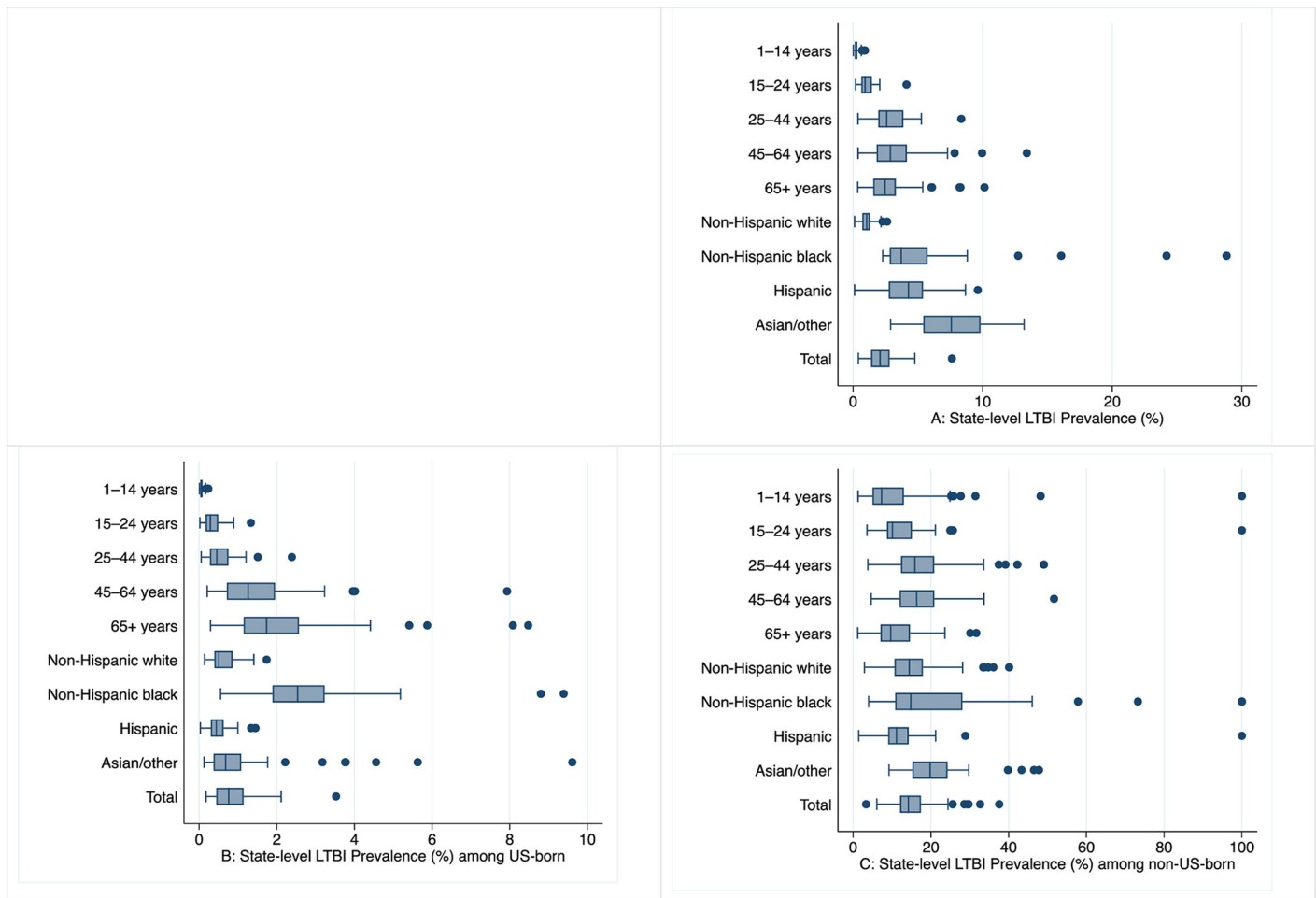

**Fig 2. Distribution of estimated state-level prevalence of latent tuberculosis infection (LTBI), within age group and race/ethnicity populations, in 50 U.S. states and District of Columbia, 2015.** Box plot A shows total population LTBI estimates for each state. Box plot B shows LTBI prevalence estimate among US-born persons. Box plot C shows LTBI prevalence estimate among non-US–born persons. Dots represent the states with outlier estimates for LTBI prevalence.

(3.1%). Among non-US–born persons, the highest LTBI prevalence was estimated in persons aged 45–64 years (16.3%) and persons of Asian or other race/ethnicity (19.1%).

Our national estimate of untreated LTBI prevalence was similar to the one reported by Haddad *et al.* (i.e., 8.9 million persons) [9]. Similar to nationally representative survey results, our results indicated substantially lower LTBI prevalence in the US-born population than in the non-US–born population of the United States; however, this ratio varied from state to state. Our results also estimated variability of LTBI prevalence among populations. This variability based on nativity was driven by both demographic differences among TB cases and previously estimated differential reactivation [14, 19] between US-born and non-US–born persons living with LTBI. Geographic variations in demographic and risk factors of TB cases can explain much of the differences in estimated LTBI prevalence seen at the state level.

Compared to the LTBI prevalence estimates from NHANES 2011–12 (4.7% total, 1.5% US-born, 20.5% non-US-born) [8], our results were much lower for total (2.7%) and non-US-born (13.9%). The estimates from NHANES were based on Tuberculin skin test (TST) results and thus might have overestimated true LTBI prevalence due to cross reaction with BCG vaccine. When Interferon Gamma Release Assay (IGRA) positivity was used in NHANES, the LTBI

estimate for non-US-born dropped to 15.9% (95%CI 13.5–18.7) [8], much closer to our estimate. However, our estimates for LTBI among older non-US-born groups was higher than NHANES IGRA findings: age 45–64 years (23.5% vs. 16.3%) and 65+ years (32.1% vs 9.8%). This may reflect cohort effects, with 2010 older birth cohorts (with higher TB infection rates) being replaced in recent years with lower infection rate younger birth cohorts [20].

Our findings suggest that one strategy to achieve TB elimination in the United States would be to prioritize all non-US–born persons, irrespective of medical risk factors, for LTBI screening and treatment. Focusing on non-US–born persons aged 25–64 years would reach up to 82.2%, or focusing on non-US-born Hispanic, Asian, or other race/ethnicity would reach up to 71.6%, of all estimated untreated LTBI among non-US–born persons. The U.S. Preventive Services Task Force guidance has recommended some components of this approach, focusing on screening asymptomatic adults born outside the United States in high TB prevalence countries and persons, regardless of nativity, living in congregate settings including correctional institutions and homeless shelters [21]. Additional guidance, such as the California TB Risk Assessment tool, recommends LTBI testing and treatment for all non-US–born individuals, individuals with immunosuppressive conditions or taking immunosuppressive therapy, and individuals who have had contact with someone with infectious TB disease during their lifetime [22]. Recent modeling has demonstrated that adherence to such approaches could substantially reduce the burden of TB disease, reducing incidence by 40% [23].

Both as a count and a proportion, the US-born population has a lower total LTBI prevalence. In addition, most of the cases attributed to recent TB transmission in the United States occur among US-born persons [2, 3, 13]. Some of our estimates of LTBI prevalence in various demographic (e.g., Hispanic persons) and medical risk groups (e.g., diabetes) are lower than those reported elsewhere [8]. The Shea *et al.* [14] difference in the reactivation rates between US-born Hispanic persons (0.178 per 100 person-years) and non-US–born Hispanic persons (0.086 per 100 person-years) applied in our back-calculation might have led to an underestimation of LTBI prevalence in Hispanic populations in comparison to the estimates from the National Health and Nutrition Examination Survey (NHANES) 2011–2012 [8]. The confidence interval for our estimate of the number of people with LTBI who also have diabetes (0.9 million, 95% CI 0.7 to 1.3) overlaps with the confidence interval reported using NHANES 2011–2012 (2.0 million, 95% CI 1.2 to 3.1) [8]. The discrepancy in point estimates may be a consequence of our assumption that all TB cases reported with no indication of diabetes status did not have diabetes, whereas all NHANES participants aged ≥12 years were systematically screened for diabetes. About 21.4% of all U.S. adults who met laboratory criteria for diabetes are not diagnosed with diabetes [24], and so it is likely that we under ascertained diabetes among TB cases.

Our study had additional limitations. First, published estimates of reactivation rates are scarce, and the 95% CIs presented in this analysis reflect only the imprecision in the input parameters derived from the Shea *et al.* and Yeats data sources [14, 15], excluding any other potential sources of uncertainty in our LTBI prevalence estimates. Second, for TB cases with more than one medical risk factor, we considered only the risk factor with the higher reactivation rate. This hierarchy might have led to an overestimation of the LTBI prevalence by medical risk factor if actually having multiple risk factors would cause an even higher reactivation risk. This limitation also prevented us from being able to provide more refined LTBI prevalence estimates for persons with multiple medical comorbidities. Third, in deriving our reactivation rates for persons with medical risk factors for progression to TB disease (**S1 Appendix in** S1 File), we assumed that the TB reactivation rates based on the total population [14, 15] could represent the experience of the population without identified medical risk factors. Fourth, time since the initial *M. tuberculosis* infection and the comorbid conditions (e.g.,

diabetes), including which occurred first, is unknown. Fifth, undocumented comorbid conditions among persons with reported TB may have led to an overestimation of LTBI in the total population, in that persons with unrecognized conditions would have been assigned lower reactivation rates than their conditions might actually engender. Sixth, we used a well-defined method [13] to distinguish cases attributed to non-recent transmission from those attributed to recent transmission of TB infection, but definitive classification using this method can be difficult. Seventh, the analysis here did not assess the effect of LTBI treatment; however, it is likely that treated LTBI would not have reactivated to TB disease. Finally, we imputed the missing data, including many cases missing data on recent transmission, which narrowed the estimated confidence limits.

Our back-calculation method has two advantages. First, our method is based on TB cases reported to the National Tuberculosis Surveillance System during 2013–2017; this high-quality dataset has standardized reporting for each case of TB disease in every U.S. state [2, 13]. These data are available, and the analysis coding has also been made publicly available, making replication and updating of LTBI prevalence estimates feasible, either by CDC or other entities. Second, our estimates describe LTBI prevalence within geographic populations as defined by age group and race/ethnicity, which combined with American Community Survey denominators can be informative for identifying those populations in the United States who would most benefit from interventions to prevent future TB cases.

## Supporting information

**S1 File.**
(DOCX)

**S1 Table. Estimated prevalence and number of persons living with latent tuberculosis infection by medical risk factor, age group, and race/ethnicity groupings, stratified by nativity and in total population, in 50 US states and District of Columbia, 2015.**
(XLSX)

## Author Contributions

**Conceptualization:** Ali Mirzazadeh, James G. Kahn, Andrew N. Hill, Suzanne M. Marks, Adam Readhead, Pennan M. Barry, Jennifer Flood, Jonathan H. Mermin, Priya B. Shete.

**Data curation:** Ali Mirzazadeh, Maryam B. Haddad, Andrew N. Hill, Suzanne M. Marks, Adam Readhead, Pennan M. Barry, Jennifer Flood, Jonathan H. Mermin, Priya B. Shete.

**Formal analysis:** Ali Mirzazadeh, Maryam B. Haddad, Andrew N. Hill.

**Funding acquisition:** Ali Mirzazadeh, James G. Kahn, Suzanne M. Marks, Priya B. Shete.

**Investigation:** Andrew N. Hill, Priya B. Shete.

**Methodology:** Ali Mirzazadeh, James G. Kahn, Maryam B. Haddad, Andrew N. Hill, Pennan M. Barry, Jennifer Flood, Jonathan H. Mermin.

**Project administration:** Maryam B. Haddad, Andrew N. Hill, Suzanne M. Marks, Priya B. Shete.

**Resources:** James G. Kahn, Jonathan H. Mermin, Priya B. Shete.

**Software:** Ali Mirzazadeh, Maryam B. Haddad, Andrew N. Hill.

**Supervision:** James G. Kahn, Andrew N. Hill, Suzanne M. Marks, Priya B. Shete.

**Validation:** Ali Mirzazadeh, Andrew N. Hill, Adam Readhead, Pennan M. Barry, Jennifer Flood.

**Visualization:** Ali Mirzazadeh, Maryam B. Haddad.

**Writing – original draft:** Ali Mirzazadeh, Maryam B. Haddad, Andrew N. Hill, Suzanne M. Marks.

**Writing – review & editing:** Ali Mirzazadeh, James G. Kahn, Maryam B. Haddad, Andrew N. Hill, Suzanne M. Marks, Adam Readhead, Pennan M. Barry, Jennifer Flood, Jonathan H. Mermin, Priya B. Shete.

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
