## [Decision Letter · Decision Letter 0]

6 Jan 2021

PONE-D-20-36817

State-level prevalence estimates of latent tuberculosis infection in the United States by medical risk factors, demographic characteristics and nativity

PLOS ONE

Dear Dr. Mirzazadeh,

Thank you for submitting your manuscript to PLOS ONE. After careful consideration, we feel that it has merit but does not fully meet PLOS ONE’s publication criteria as it currently stands. Therefore, we invite you to submit a revised version of the manuscript that addresses the points raised during the review process.

Please submit your revised manuscript. If you will need significantly more time to complete your revisions, please reply to this message or contact the journal office at plosone@plos.org. Please include the following items when submitting your revised manuscript:

We look forward to receiving your revised manuscript.

Kind regards,

Frederick Quinn

Academic Editor

PLOS ONE

Journal Requirements:

3. We note that Figure 1 and 2 and Supporting Information Figures S1 and S2 in your submission contain map images which may be copyrighted.

a. You may seek permission from the original copyright holder of Figure 1 and 2 and Supporting Information Figures S1 and S2 to publish the content specifically under the CC BY 4.0 license. 

b, If you are unable to obtain permission from the original copyright holder to publish these figures under the CC BY 4.0 license or if the copyright holder’s requirements are incompatible with the CC BY 4.0 license, please either i) remove the figure or ii) supply a replacement figure that complies with the CC BY 4.0 license. Please check copyright information on all replacement figures and update the figure caption with source information. If applicable, please specify in the figure caption text when a figure is similar but not identical to the original image and is therefore for illustrative purposes only.

Reviewers' comments:

Reviewer's Responses to Questions

**Comments to the Author**

1. Is the manuscript technically sound, and do the data support the conclusions?

Reviewer #1: Yes

Reviewer #2: Yes

2. Has the statistical analysis been performed appropriately and rigorously? 

Reviewer #1: Yes

Reviewer #2: Yes

3. Have the authors made all data underlying the findings in their manuscript fully available?

Reviewer #1: Yes

Reviewer #2: Yes

4. Is the manuscript presented in an intelligible fashion and written in standard English?

Reviewer #1: Yes

Reviewer #2: Yes

5. Review Comments to the Author

Reviewer #1: PONE-D-20-36817

“State-level prevalence estimates of latent tuberculosis infection in the United States by medical risk factors, demographic characteristics and nativity”

Summary: This manuscript used previously published TB reactivation rates from the literature and back-calculated active tuberculosis cases to estimate current LTBI prevalence in the US. They examined these prevalences by age, race/ethnicity, a few selected comorbid conditions and nativity.

Major comments:

1. I think the method of trying to use active TB cases to back calculate LTBI prevalence is interesting though this manuscript estimated 8.6 million of Americans having LTBI and this is substantially lower than what NHANES survey estimated (13.3 million). Where do you think the large discrepancy is coming from?

2. I think this manuscript and the data and results presented are intriguing. However, it is adding only a few additional components with limitations to the analyses by Shea et al and Haddad et al. For example, this analysis included comorbid conditions but significant assumptions had to be made when deriving the prevalence of comorbid conditions (from the active TB cases) including whether TB came before the comorbid conditions. Also, the model doesn’t take into account treatment for LTBI and if you are arguing that these estimates could be use to target testing and treating, this is big point.

3. I think generally there is agreement that you should screen those who were not born in the US for LTBI and then consider treatment. I am not sure that this manuscript adds evidence one way or another to the treatment question.

4. I think it is admirable to include the other comorbidities but they are included as only what comorbidity at a time. Also the immunosuppressive therapy category is huge but what about the disease modifying agents for all of the rheumatologic conditions? Also what about malignancies?

5. I wonder if you could include treatment with isoniazid into the model to try and refine the estimates and include some of the treatment for LTBI.

Minor comments:

1. I’m having a hard time reconciling that person of Asian or other race/ethnicity in the total US population had the highest LTBI prevalence, but then in the next paragraph it said that most of the persons estimated as having LTBI were non-Hispanic white.

2. When describing the state prevalence of LTBI, I’m not sure how informative describe LTBI prevalence >1% and then in 26 states it was over 15% in non-US born. Are there reasons for describing these cut-offs?

Reviewer #2: The manuscript entitled “State-level prevalence estimates of latent tuberculosis infection in the United States by medical risk factors, demographic characteristics and nativity”. Overall, the manuscript is well organized.

General comments

The study is interesting and gives good figure on LTBI status in US. However, I have the following concerns.

1. The study is talking about prevalence of LTBI in 2015. Now it is 2020. What is the relevant of your study on today epidemiology as well as prevention and control TB in US?

2. As you mentioned in your rationale, study on prevalence of LTBI in US was conducted from 2011 to 2015 by Haddad et al. In addition, in 2nd paragraph of your discussion you mentioned as LTBI national survey was conducted in US and high prevalence of LTBI was reported among non-US born. Moreover, in the same paragraph you mentioned high LTBI in California. Concerning these, again what is the relevance of your study to US?

Specific comments

1. In result part under sub-title characteristics of personals predicted to have LTBI: Remove the word few from the initial of two paragraphs because even the percentages are low the actual numbers are more than hundred thousand.

2. You mentioned about seven limitations and three advantages of your study. How you comment on this? On the top, one of your advantages was having similar result with the previous two studies in the area. How come having similar finding become advantage of your study?

Final decision: The manuscript is accepted for publication with minor revision

6. PLOS authors have the option to publish the peer review history of their article (what does this mean?). If published, this will include your full peer review and any attached files.

Reviewer #1: No

Reviewer #2: No

---

## [Author Response · Author response to Decision Letter 0]

21 Feb 2021

Please see the Word Doc we attached. Thank you!

---

## [Decision Letter · Decision Letter 1]

10 Mar 2021

State-level prevalence estimates of latent tuberculosis infection in the United States by medical risk factors, demographic characteristics and nativity

PONE-D-20-36817R1

Dear Dr. Mirzazadeh,

We’re pleased to inform you that your manuscript has been judged scientifically suitable for publication and will be formally accepted for publication once it meets all outstanding technical requirements.

Kind regards,

Frederick Quinn

Academic Editor

PLOS ONE

Additional Editor Comments (optional):

Reviewers' comments:

Reviewer's Responses to Questions

**Comments to the Author**

1. If the authors have adequately addressed your comments raised in a previous round of review and you feel that this manuscript is now acceptable for publication, you may indicate that here to bypass the “Comments to the Author” section, enter your conflict of interest statement in the “Confidential to Editor” section, and submit your "Accept" recommendation.

Reviewer #1: All comments have been addressed

Reviewer #2: All comments have been addressed

2. Is the manuscript technically sound, and do the data support the conclusions?

Reviewer #1: Yes

Reviewer #2: Yes

3. Has the statistical analysis been performed appropriately and rigorously? 

Reviewer #1: Yes

Reviewer #2: Yes

4. Have the authors made all data underlying the findings in their manuscript fully available?

Reviewer #1: Yes

Reviewer #2: Yes

5. Is the manuscript presented in an intelligible fashion and written in standard English?

Reviewer #1: Yes

Reviewer #2: Yes

6. Review Comments to the Author

Reviewer #1: (No Response)

Reviewer #2: I checked and all my comments are incorporated. I approved for publication.

7. PLOS authors have the option to publish the peer review history of their article (what does this mean?). If published, this will include your full peer review and any attached files.

Reviewer #1: No

Reviewer #2: No

---

## [Editor Report · Acceptance letter]

18 Mar 2021

PONE-D-20-36817R1 

State-level prevalence estimates of latent tuberculosis infection in the United States by medical risk factors, demographic characteristics and nativity 

Dear Dr. Mirzazadeh:

I'm pleased to inform you that your manuscript has been deemed suitable for publication in PLOS ONE. Congratulations! Your manuscript is now with our production department. 

Kind regards, 

on behalf of

Dr. Frederick Quinn 

Academic Editor

PLOS ONE